# Backpropagation at the Infinitesimal Inference Limit of Energy-Based Models: Unifying Predictive Coding, Equilibrium Propagation, and Contrastive Hebbian Learning

**Beren Millidge**
MRC Brain Network Dynamics Unit
University of Oxford, UK
beren@millidge.name

**Yuhang Song** *
MRC Brain Network Dynamics Unit
University of Oxford, UK
yuhang.song@ndcn.ox.ac.uk

**Tommaso Salvatori**
Institute of Logic and Computation
TU Wien, Austria
tommaso.salvatori@cs.ox.ac.uk

**Thomas Lukasiewicz**
TU Wien, Austria
University of Oxford, UK
thomas.lukasiewicz@tuwien.ac.at

**Rafal Bogacz**
MRC Brain Network Dynamics Unit
University of Oxford, UK
rafal.bogacz@ndcn.ox.ac.uk

## Abstract

How the brain performs credit assignment is a fundamental unsolved problem in neuroscience. Many 'biologically plausible' algorithms have been proposed, which compute gradients that approximate those computed by backpropagation (BP), and which operate in ways that more closely satisfy the constraints imposed by neural circuitry. Many such algorithms utilize the framework of energy-based models (EBMs), in which all free variables in the model are optimized to minimize a global energy function. However, in the literature, these algorithms exist in isolation and no unified theory exists linking them together. Here, we provide a comprehensive theory of the conditions under which EBMs can approximate BP, which lets us unify many of the BP approximation results in the literature (namely, predictive coding, equilibrium propagation, and contrastive Hebbian learning) and demonstrate that their approximation to BP arises from a simple and general mathematical property of EBMs at free-phase equilibrium. This property can then be exploited in different ways with different energy functions, and these specific choices yield a family of BP-approximating algorithms, which both includes the known results in the literature and can be used to derive new ones.

## 1 Introduction

The backpropagation of error algorithm (BP) (Rumelhart et al., 1986) has become the workhorse algorithm underlying the recent successes of deep learning (Krizhevsky et al., 2012; Silver et al., 2016; Vaswani et al., 2017). However, from a neuroscientific perspective, BP has often been criticised as not being biologically plausible (Crick et al., 1989; Stork, 1989). Given that the brain faces a credit assignment problem at least as challenging as deep neural networks, there is a fundamental question of whether the brain uses backpropagation to perform credit assignment. The answer to this question depends on whether there exist biologically plausible algorithms which approximate BP that could be implemented in neural circuitry (Whittington & Bogacz, 2019; Lillicrap et al., 2020). A large number of potential algorithms have been proposed in the literature (Lillicrap et al., 2016; Xie & Seung, 2003; Nøkland, 2016; Whittington & Bogacz, 2017; Lee et al., 2015; Bengio & Fischer, 2015; Millidge et al., 2020c;a; Song et al., 2020; Ororbia & Mali, 2019), however insight into the linkages and relationships between them is scarce, and thus far the field largely presents itself as a set of disparate algorithms and ideas without any unifying or fundamental principles.

In this paper we provide a theoretical framework which unifies four disparate schemes for approximating BP – predictive coding with weak feedback (Whittington & Bogacz, 2017) and on the first step after initialization (Song et al., 2020), the Equilibrium Propagation (EP) framework (Scellier & Bengio, 2017), and Contrastive Hebbian Learning (CHL) (Xie & Seung, 2003). We show that these algorithms all emerge as special cases of a general mathematical property of the energy based

---

*Corresponding author.

model framework that underlies them and, as such, can be generalized to novel energy functions and to derive novel algorithms that have not yet been described in the literature.

The key insight is that for energy based models (EBMs) underlying these algorithms, the total energy can be decomposed into a component corresponding to the supervised loss function that depends on the output of the network, and a second component that relates to the 'internal energy' of the network. Crucially, at the minimum of the internal energy, the dynamics of the neurons point exactly in the direction of the gradient of the supervised loss. Thus, at this point, the network dynamics implicitly provide an exact gradient signal. This fact can be exploited in two ways. Firstly, this instantaneous direction can be directly extracted and used to perform weight updates. This 'first step' approach is taken in (Song et al., 2020) and results in exact BP using only the intrinsic dynamics of the EBM, but typically requires a complex set of control signals to specify when updates should occur. Alternatively, a second equilibrium can be found close to the initial one. If it is close enough – a condition we call the *infinitesimal inference limit* –, then the vector from the initial to the second equilibrium approximates the direction of the initial dynamics and thus the difference in equilibria approximates the BP loss gradient. This fact is then utilized either implicitly (Whittington & Bogacz, 2017) or explicitly (Scellier & Bengio, 2017; Xie & Seung, 2003) to derive algorithms which approximate BP. Once at this equilibrium, all weights can be updated in parallel rather than sequentially and no control signals are needed.

The paper is structured as follows. First, we provided concise introductions to predictive coding networks (PCNs), contrastive Hebbian learning (CHL), and equilibrium propagation (EP). Then, we derive our fundamental results on EBMs, show how this results in a unified framework for understanding existing algorithms, and showcase how to generalize our framework to derive novel BP-approximating algorithms.

## 2 BACKGROUND AND NOTATION

We assume we are performing credit assignment on a hierarchical stack of feedforward layers $x_0 \ldots x_L$ with layer indices $0 \ldots L$ in a supervised learning task. The input layer $x_0$ is fixed to some data element $d$. The layer states $x_l$ are vectors of real values and represent a rate-coded average firing rate for each neuron. We assume a supervised learning paradigm in which a target vector $T$ is provided to the output layer of the network and the network as a whole minimizes a supervised loss $\mathcal{L}(x_L, T)$. $T$ is usually assumed to be a one-hot vector in classification tasks but this is not necessary. Synaptic weight matrices $W_l$ at each layer affect the dynamics of the layer above. The network states $x_l$ are assumed to be free parameters that can vary during inference. We use $x = \{x_l\}$ and $W = \{W_l\}$ to refer to the sets across all layers when specific layer indices are not important.

The models we describe are EBMs, which possess an energy function $E(x_0 \ldots x_L, W_0 \ldots W_L, T)$ which is minimized both with respect to neural activities $x_l$ and weights $W_l$ with dynamics,

$$\text{Inference:} \quad \frac{dx_l}{dt} = -\frac{\partial E}{\partial x_l} \tag{1}$$

$$\text{Learning:} \quad \frac{dW_l}{dt} = -\frac{\partial E}{\partial W_l} \tag{2}$$

which are simply a gradient descent on the energy. For notational simplicity we implicitly set the learning rate $\eta = 1$ throughout. Moreover, we assume differentiability (at least up to second order) of the energy function with respect to both weights and activities. In EBMs, we run 'inference' to optimize the activities first using Equation 1 until convergence. Then, after the inference phase is complete we run a single step of weight updates according to Equation 2, which is often called the learning phase. A schematic representation of how EBMs learn with a global energy function is presented in Figure 1A while a visual breakdown of our results is presented in Figure 1B.

### 2.1 PREDICTIVE CODING

Predictive Coding (PC) emerged as a theory of neural processing in the retina (Srinivasan et al., 1982) and was extended to a general theory of cortical function (Mumford, 1992; Rao & Ballard, 1999; Friston, 2005). The fundamental idea is that the brain performs inference and learning by learning to predict its sensory stimulu and minimizing the resulting *prediction errors*. Such an approach provides a natural unsupervised learning objective for the brain (Rao & Ballard, 1999), while also minimizing redundancy and maximizing information transmission by transmitting only unpredicted information (Barlow, 1961; Sterling & Laughlin, 2015). The learning rules used by PCNs require only local and Hebbian updates (Millidge et al., 2020b) and a variety of neural microcircuits have been proposed that can implement the computations required by PC (Bastos et al., 2012; Keller & Mrsic-Flogel, 2018). Moreover, recent works have begun exploring the use of large-scale PCNs in machine learning tasks, to some success (Millidge, 2019; Salvatori et al., 2021;

2022; Kinghorn et al., 2021; Millidge et al., 2022; Lotter et al., 2016; Ofner & Stober, 2021). Unlike the other algorithms presented here, PC has a mathematical interpretation as in terms of variational Bayesian inference (Friston, 2005; 2003; Bogacz, 2017; Buckley et al., 2017; Millidge et al., 2021) and the variables in the model can be mapped to explicit probabilistic elements of a generative model. The energy function minimized by a PCN is written as $E_{\text{PC}} = \sum_{l=1}^{L} \epsilon_l^2$, where the $\epsilon_l = \Pi_l(x_l - f(W_{l-1}x_{l-1}))$ terms are prediction errors since they denote the difference between the activity $x_l$ of a layer and the 'prediction' of that activity from the layer below $f(W_l x_{l-1})$. The prediction error terms are weighted by their inverse covariance $\Pi_l$, and this provides the network with a degree of knowledge about its own uncertainty. The ratio of these precisions $\frac{\Pi_l}{\Pi_{l-1}}$ effectively controls the weighting of bottom-up vs top-down information flow through a PCN. Given this energy function, the dynamics of the PCN can be derived from the general EBM dynamics above as follows,

$$\frac{dx_l}{dt} = -\frac{\partial E_{\text{PC}}}{\partial x_l} = \Pi_l \epsilon_l - \Pi_{l+1}\epsilon_{l+1}\frac{\partial f(W_l x_l)}{\partial x_l} \tag{3}$$

$$\frac{dW_l}{dt} = -\frac{\partial E_{\text{PC}}}{\partial W_l} = \Pi_l \epsilon_{l+1}\frac{\partial f(W_l x_l)}{\partial W_l} \tag{4}$$

To simplify notation, for most derivations we assume identity precision matrices $\Pi_l = \Pi_{l+1} = I$. Several limits in which PC approximates BP have been observed in the literature (Whittington & Bogacz, 2017; Millidge et al., 2020a; Song et al., 2020). We show that the results of (Whittington & Bogacz, 2017) and (Song et al., 2020) can be explained through our general framework, since both rely on properties of the infinitesimal inference limit. Specifically, (Whittington & Bogacz, 2017) demonstrate convergence to BP as the precision ratio is heavily weighted towards bottom up information. Secondly (Song et al., 2020) take advantage of the dynamics of the EBM at equilibrium to devise an PCN that implements exact BP.

## 2.2 Contrastive Hebbian Learning

Contrastive Hebbian Learning (CHL) originated as a way to train continuous Hopfield networks (Hopfield, 1984) without the pathologies that arise with pure Hebbian learning (Hebb, 1949). Because pure Hebbian learning simply strengthens synapses that fire together, it induces a positive feedback loop which eventually drives many weights to infinity. CHL approaches handle this by splitting learning into two phases (Galland & Hinton, 1991; Movellan, 1991). In the first phase (free phase), the network output is left to vary freely while the input is fixed, and the network state converges to an equilibrium known as the free phase equilibrium $\bar{x}$. At the free phase equilibrium, a standard Hebbian weight update is performed $\Delta W \propto \bar{x}\bar{x}^T$. Then in the second phase, the network output is clamped to the desired targets $T$ and the network converges to a new equilibrium: the clamped phase equilibrium $\tilde{x}$. The weights are then updated with a negative anti-Hebbian update rule $\Delta W \propto -\tilde{x}\tilde{x}^T$. Instead of updating weights individually at each phase, one can instead write a single weight update rule that occurs at the end of the clamped phase,

$$\Delta W_{\text{CHL}} = -(\tilde{x}\tilde{x}^T - \bar{x}\bar{x}^T) \tag{5}$$

The intuition is that neurons which are co-active at the clamped phase have their synaptic connections strengthened while those that are co-active in the free phase are weakened, thus strengthening connections that are in some sense involved in correctly responding to the target while weakening all others. Mathematically, CHL performs a gradient descent on a contrastive loss function,

$$C(W) = E^{\tilde{x}}(\tilde{x}, W) - E^{\bar{x}}(\bar{x}, W) \tag{6}$$

where the individual energies $E^{\tilde{x}}$ and $E^{\bar{x}}$ refer to the standard Hopfield energy (Hopfield, 1982) at either the free phase or clamped phase equilibria. The Hopfield energy is defined as,

$$E_{\text{CHL}}(x, W) = x^T W x + x^T b \tag{7}$$

where $b$ is a bias vector. In one of the first results demonstrating an approximation to BP, (Xie & Seung, 2003) showed that in the limit of geometrically decreasing feedback from higher to lower layers, the updates of CHL approximate BP to first order.

## 2.3 Equilibrium Propagation

Equilibrium propagation (EP) (Scellier & Bengio, 2017) can be considered a contrastive Hebbian method based on an infinitesimal perturbation of the loss function. Instead of clamping the output of

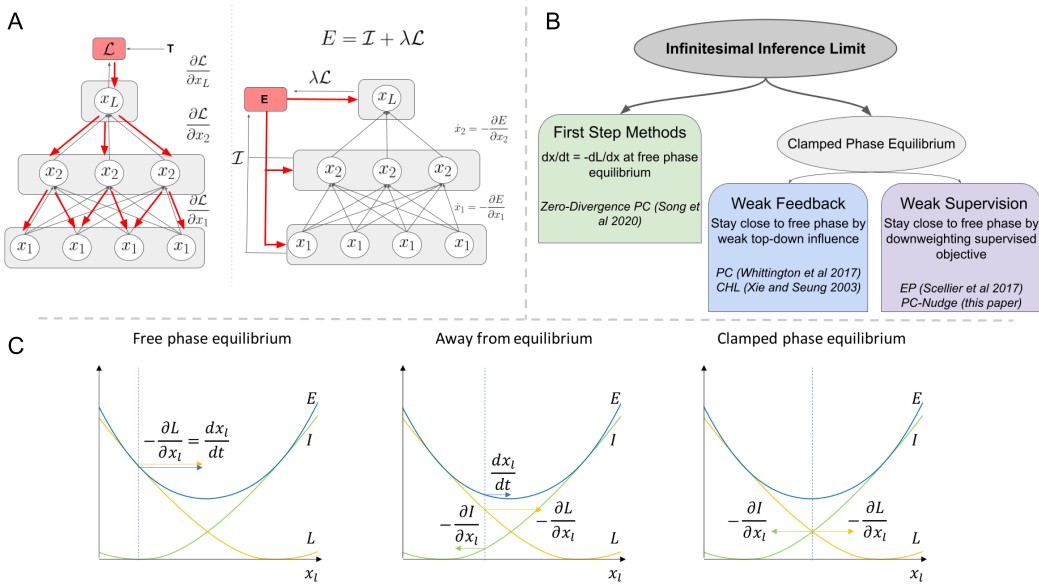

Figure 1: **A**: An ANN optimizing a supervised loss function with BP vs an EBM. For the ANN, the supervised loss is only affected by the output, and gradients are sequentially propagated backwards through the network. For the EBM, the energy is a global function of all variables, and all variables update themselves to minimize the energy according to Equation 1. **B**: The hierarchy of methods approximating BP. Exactly at the free phase, the dynamics point in the direction of the supervised loss gradient. First-step algorithms exploit this fact to perform exact backprop. If the network is run to a clamped phase equilibrium, the gradients only converge to BP as the clamped phase converges to the free phase equilibrium, which we call the infinitesimal inference limit. This can be assured either by ensuring that top-down information propagation through the network is weak (the weak feedback limit) or else ensuring that the supervised loss has only a small effect on the total energy (weak supervision limit). **C**: A schematic visualization of the infinitesimal inference limit. Away from equilibrium, the dynamics are determined by contributions from both the internal energy and the supervised loss. At the free phase equilibrium, the contribution from the internal energy is 0 so the dynamics follow exactly the gradient of the supervised loss. At the clamped phase equilibrium, the contributions of the internal and supervised losses perfectly cancel each other out.

the network to the target, the outputs are clamped to a 'nudged' version of the input where the targets only possess a small influence on the output. This influence is parametrized by a scalar parameter $\lambda$. Like in CHL the network operates in two phases – a free phase and a 'nudged' phase. In the free phase, the network minimizes an energy function $E_{\text{EP}}^{\lambda=0}$ which is usually the Hopfield energy. In the 'nudged phase', the network instead optimizes a nudged energy function $E_{\text{EP}}^{\lambda} = E_{\text{EP}} + \lambda\mathcal{L}$ where $L$ is the supervised loss function at the output layer and $0 \leq \lambda \leq 1$ is a scalar parameter which is assumed to be small. The activities at the free phase equilibrium, as in CHL, are denoted by $\bar{x}$ and the clamped phase equilibrium as $\tilde{x}$. EP then updates the weights with the following rule,

$$\Delta W = \frac{1}{\lambda}\Big[\frac{\partial E_{\text{EP}}^{\lambda}}{\partial W}(\tilde{x}, W) - \frac{\partial E_{\text{EP}}^{\lambda=0}}{\partial W}(\bar{x}, W)\Big] \tag{8}$$

which is the difference between the weight gradients at the nudged phase equilibrium and the free phase equilibrium. It was proven in (Scellier & Bengio, 2017) that in the limit as $\lambda \to 0$, the EP update rule converges to BP as long as the energy gradients are assumed to be differentiable. Beyond this, it was shown in (Scellier & Bengio, 2019) that in a RNN with static input the dynamics of the weight updates in EP are identical to that of recurrent BP (Almeida, 1990; Pineda, 1987). Recently, there has been significant work scaling up EP to train large scale neural networks with significant success (Laborieux et al., 2021; Ernoult et al., 2020) as well as implementations using analog hardware (Kendall et al., 2020) which utilizes the ability to implement the inference dynamics directly through physics to achieve significant computational speedups.

## 3 THEORETICAL RESULTS

### 3.1 GENERAL ENERGY BASED MODEL

Here, we demonstrate how these disparate algorithms and their relationship with BP can all be derived from a basic mathematical property of EBMs. A key fact about the EBMs utilized in these

algorithms is that the total energy $E$ can be decomposed into a sum of two components. An energy function at the output layer which corresponds to the supervised loss $\mathcal{L}$, and an 'internal energy' function $\mathcal{I}$ which corresponds to the energy of all the layers in the network except the output layer. Typically, the internal energy can be further subdivided into layerwise local energies for each layer $\mathcal{I} = \sum_{l=1}^{L-1} E_l$. If we add a scaling coefficient $0 \leq \lambda \leq 1$ to scale the relative contributions of the internal and supervised losses, we can write the total energy as,

$$E = \mathcal{I} + \lambda \mathcal{L} \tag{9}$$

The EBM is then run in two phases. In the free phase, the output is left to vary freely, so that the targets have no impact upon the dynamics. This is equivalent to ignoring the supervised loss and hence setting $\lambda = 0$. In the clamped (or 'nudged') phase, the output layer is either clamped or nudged towards the targets. This means that the targets do have an impact on the dynamics and thus $\lambda > 0$. We now consider the dynamics of the network at the equilibrium at the free phase. Since the dynamics of the activities of the network are simply a gradient descent on the energy, we have that,

$$\frac{dx}{dt}\bigg|_{\text{free phase}} = 0 \implies \frac{\partial \mathcal{I}}{\partial x} = 0 \tag{10}$$

Since, during the free phase $\lambda = 0$ and so $E = \mathcal{I}$. If we then begin the clamped phase, but start out at the free phase equilibrium of the activities, it is straightforward to see that,

$$\frac{dx}{dt}\bigg|_{\text{clamped phase}} = -\frac{\partial E}{\partial x} = -\frac{\partial \mathcal{I}}{\partial x} - \lambda \frac{\partial \mathcal{L}}{\partial x}$$
$$\frac{\partial \mathcal{I}}{\partial x} = 0 \implies \frac{dx}{dt} = -\lambda \frac{\partial \mathcal{L}}{\partial x} \tag{11}$$

and thus that, when initialized at the free phase equilibrium, the activity dynamics in the clamped phase follow the negative gradient of the BP loss. As a mathematical consequence of the decomposition of the energy function, at the free phase equilibrium, information about the supervised loss is implicitly backpropagated to all layers of the network to determine their dynamics.

An intuitive way to view this, which is presented in Figure 1C, is that during inference, we can interpret the activity dynamics during the clamped phase as balancing between two forces – one pushing to minimize the internal energy and one pushing to minimize the supervised loss. At the minimum of the internal energy (which is the equilibrium of the free phase), the 'force' of the internal energy is $0$ and so the only remaining force is that corresponding to the gradient of the supervised loss. Moreover, as long as the activities remain 'close enough' to the free phase equilibrium, the dynamics should still be dominated by the supervised loss and thus the activity dynamics closely approximate the BP loss gradients. We call the limit where the clamped phase equilibrium converges to the free phase equilibrium the *infinitesimal inference limit*, since it is effectively stating that the 'inference phase' of the network is only achieving an infinitesimal change to the initial (free phase) 'beliefs'. This is the general condition which unifies the disparate BP-approximating algorithms from the literature which differ primarily in how they attain the infinitesimal inference limit.

Empirically, we investigate this condition in Figure 2. In panel A we plot the evolution of the three energy components during the clamped phase in a randomly initialized 4-layer nonlinear EBM with relu activation functions. We can see that, by starting at the free phase, the internal energy component is initialized at 0 and hence at the initial step, the slope of the total energy and that of the backprop loss are identical which is the key relationship that underpins the infinitesimal inference limit. In panel B, we quantify this effect and show that the distance between the change in neural activities and the backprop gradients increases rapidly with the number of inference steps taken until the distance saturates at the clamped equilibrium some distance away from the free equilibrium.

It is important to note that in layerwise architectures initialized at the free phase equilibrium, information has to propagate back in a layerwise fashion from the clamped output. This means that only the penultimate layer is affected first, and then one layer down is perturbed at each timestep. What this means is that the 'first' motion of the activities in a layer in the clamped phase are guaranteed to minimize the BP loss – not that all these motions occur simultaneously. This is the property that is exploited in the single step algorithms below. If, instead, it is required to update all weights simultaneously, instead of in a layerwise fashion, it is necessary to wait for information to propagate through the entire network and ultimately converge to the clamped phase fixed point. It is this requirement that leads to the fixed point algorithms in CHL, EP, and PC which, since more than one step is taken away from the free phase fixed point, results in approximate BP instead.

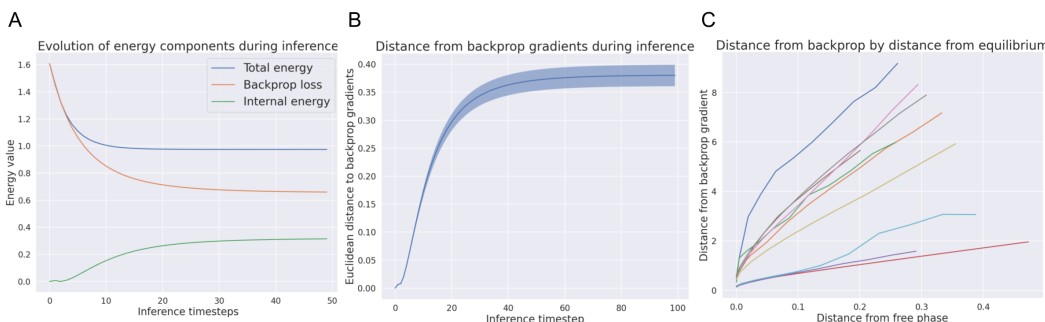

Figure 2: **A**: The evolution of the energies during an inference phase starting at the free phase. The supervised loss rapidly decreases while the internal energy increases (from 0). The total energy declines as the increase in internal energy is counteracted by a larger decrease of the supervised loss. At the initial condition where the internal energy is 0 (free phase) the change in energy is identical to the change in the backprop loss. **B**: Euclidean distance between the backprop gradients and the change in activity values during an inference phase. We see again that at the beginning of inference, a close match to backprop is obtained while if inference continues for a long time this match decreases. **C**: Relationships between distance between the clamped and free phase equilibria and the distance between the estimated gradients and the true BP gradients for multiple initializations of a small MLP. As predicted by the Taylor expansion, for most initializations the approximation error grows linearly with distance between equilibria although some curves appear to bend slightly, especially for larger distances, indicating the breakdown of the local linearity assumption. All results obtained in a three-layer nonlinear (relu) neural network using the predictive coding quadratic energy function.

## 3.2 FIRST STEP ALGORITHMS

A simple approach to exploit the dynamics at the free phase equilibrium is to directly use the direction of the activity dynamics at the first step to update the weights. This approach can compute the exact BP gradients since the gradients of the loss with respect to the weights can be expressed as,

$$\frac{\partial \mathcal{L}}{\partial W_l} = \frac{\partial \mathcal{L}}{\partial x_{l+1}} \frac{\partial x_{l+1}}{\partial W_l} = \frac{1}{\lambda} \frac{\partial E}{\partial x_{l+1}}\bigg|_{x_{l+1}^-} \frac{\partial x_{l+1}}{\partial W_l}$$
$$= -\frac{1}{\lambda} \frac{dx_{l+1}}{dt}\bigg|_{\bar{x}_{l+1}} \frac{\partial x_{l+1}}{\partial W_l} \tag{12}$$

A specific case of this relation in PCNs was derived by (Song et al., 2020). To understand it concretely, we specialize the derivation to the dynamics of PCNs as in (Song et al., 2020). The free phase equilibrium of PCNs is equal to the feedforward pass in the equivalent ANN $\bar{x}_l = f(W_{l-1}\bar{x}_{l-1})$, where $f$ is an activation function applied elementwise. A key property of PC is that the internal energy is 0 at the free phase equilibrium. Thus all prediction errors throughout the network are zero: $\epsilon_l = 0$. In the clamped phase, the output layer is clamped to the target and a prediction error $\epsilon_L$ is generated at the output layer. The dynamics of the activities follow Equation 3, but $\epsilon_l = 0$. This implies that after the first step away from the free phase equilibrium the activities are,

$$\frac{dx_l}{dt} = \epsilon_{l+1} \frac{\partial f(W_l x_l)}{\partial x_l} \implies x_l^{t+1} = \bar{x}_l + \epsilon_{l+1}^t \frac{\partial f(W_l x_l^t)}{\partial x_l^t} \tag{13}$$

And thus that at the first step $t+1$ after the free phase equilibrium, the prediction error $\epsilon_l$ is,

$$\epsilon_l^{t+1} = x_l^{t+1} - \bar{x}_l = \bar{x}_l + \epsilon_{l+1}^t \frac{\partial f(W_l x_l)}{\partial x_l} - \bar{x}_l = \epsilon_{l+1}^t \frac{\partial f(W_l x_l^t)}{\partial x_l^t} \tag{14}$$

However, this is simply a scaled version of the BP recursion where $\epsilon_{l+1}$ takes the role of the adjoint. This means that on the first step away from the free phase equilibrium, PCNs implement exact BP. Importantly though, this correspondence is not unique to PCNs, but emerges as a consequence of the mathematical structure of the energy function, and thus can occur in any EBM with such a structure and where the free phase equilibrium is equal to the forward pass of the network.

## 3.3 THE INFINITESIMAL INFERENCE LIMIT

Instead of utilizing the instantaneous dynamics at the free phase equilibrium, we can also use the property that the dynamics direction is approximately the same as the gradient direction as long as

the activities remain 'close' to the free phase equilibrium to derive an algorithm that approximates BP and converges to it in the limit of the free and clamped equilibria coinciding. We consider algorithms that converge to a second 'clamped' fixed point $\tilde{x}$ based on the dynamics of the total energy function with a contribution from the supervised loss – i.e., with $\lambda > 0$. Given both the clamped and free fixed points, we show that we can approximate the supervised loss gradient as the difference between the gradients at the clamped and free fixed points. If we assume that the gradients of each component of the energy with respect to the weights are continuous and differentiable, we can derive this approximation directly by considering a first order Taylor expansion of the clamped fixed point around the free fixed point $\bar{x}$,

$$
\begin{aligned}
\frac{\partial E^\lambda}{\partial W}\Big|_{x=x^*} &\approx \frac{\partial E^\lambda}{\partial W}\Big|_{x=\bar{x}} + \frac{\partial^2 E^\lambda}{\partial x \partial W}\Big|_{x=\bar{x}}(x^* - \bar{x}) + \mathcal{O}(\delta_x)^2 \\
&\approx \frac{\partial I^\lambda}{\partial W}\Big|_{x=\bar{x}} + \lambda\frac{\partial \mathcal{L}^\lambda}{\partial W}\Big|_{x=\bar{x}} + H_{\bar{x}}\delta_x + \mathcal{O}(\delta_x)^2
\end{aligned}
\tag{15}
$$

Where in the second line we have used the decomposition of the energy (Equation 9) and we have defined $\delta_x = (x^* - \bar{x})$ and $H_{\bar{x}} = \frac{\partial^2 E^\lambda}{\partial x \partial W}|_{x=\bar{x}}$. Using Equation 9 and the fact that the internal energy does not depend on $\lambda$ and thus the gradient of the internal energy at the activity values of the free phase equilibrium is the same as the gradient of the total energy at the free phase equilibrium – i.e., $\frac{\partial I^\lambda}{\partial W} = \frac{\partial E^{\lambda=0}}{\partial W}$, we can rearrange this to obtain,

$$
\lambda\frac{\partial \mathcal{L}}{\partial W}\Big|_{x=\bar{x}} \approx \frac{\partial E^\lambda}{\partial W}\Big|_{x=x^*} - \frac{\partial E^{\lambda=0}}{\partial W}\Big|_{x=\bar{x}} - H_{\bar{x}}\delta_x - \mathcal{O}(\delta_x)^2
\tag{16}
$$

which is the CHL update rule (Movellan, 1991) as an approximation to the gradient of the supervised loss at the free phase equilibrium. This provides a mathematical characterisation of the conditions for when this approximation is accurate and hence the CHL algorithm well approximates the BP gradient. A key metric for the validity of our Taylor expansion approximation is that it predicts that the relationship between the approximation error to the backprop gradient and the distance between the free and clamped equilibria should be linear, at least when the equilibria are close. This is due to the dominance of the first-order linear term in the expansion. We empirically test this in Figure 2C, and show that for a large number of initializations, this tends to be correct, at least for small distances, thus showing that the first order terms appear able to model the approximation error well.

By observing the explicit error term, we see that it depends on two quantities. Firstly, the crossderivative of the clamped phase energy at the free phase equilibrium $H_{\bar{x}}$ with respect to both activities and weights, and secondly the distance between activities at the free and clamped phase equilibrium $\delta_x$. Intuitively, the crossderivative term measures how much the weight gradient changes as the activity equilibrium moves. Thus, when this term is large, the energy is in a highly sensitive region, so that the gradients computed at a clamped phase equilibrium a small distance away from free phase equilibrium might not be as accurate.

This crossderivative term depends heavily on the definition of the energy function, as well as the location of the free phase equilibrium in the parameter space and thus this means of reducing the approximation error has largely been ignored by existing algorithms which focus primarily on reducing $\delta_x$. Our analysis does, however, suggest the possibility of 'correcting' the CHL update by subtracting out the first order taylor expansion term $H_{\bar{x}}\delta_x$. For many energy functions, this derivative may be obtained analytically or else by automatic differentiation software (Griewank et al., 1989) and $\delta_x$ is straightforward to compute. This would then make the corrected CHL update approximate BP to second order instead of the first order approximation that is currently widely used.

Reducing $\delta_x$ means ensuring that the clamped phase equilibrium is as close as possible to the free phase. This can be accomplished in two ways. The first, which is implemented in CHL, is to still clamp the output units of the network, but to ensure that feedback connections are weak so that even with output units clamped, the equilibrium does not diverge far from the free phase. This approach is taken in (Xie & Seung, 2003) who require geometrically decaying feedback strength across layers (parametrerized by a scalar parameter $0 \leq \gamma \leq 1$) to obtain a close approximation to BP. This is why we call this condition the 'weak feedback limit'. A similar approach is taken in (Whittington & Bogacz, 2017) who similarly require an geometrically decaying precision ratio across layers to obtain a close approximation to BP in PCNs. Alternatively, we can instead maintain the feedback strength but reduce the importance of the supervised loss on the network dynamics – i.e. turn a clamped output to a 'nudged' output which we call the 'weak supervision limit'. This alternate

approach is taken in the EP algorithm where an almost identical algorithm is derived by instead taking an infintesimal perturbation of the $\lambda$ parameter.

### 3.3.1 PREDICTIVE CODING AS CONTRASTIVE HEBBIAN LEARNING

Using our new framework, it is straightforward to derive PC as a CHL algorithm which simply uses the variational free energy instead of the Hopfield energy. PCNs have a unique property that the free phase equilibrium is simply the feedforward pass of the network and that all prediction errors $\epsilon_l$ in the network are zero which implies the free phase weight gradient is also zero since,

$$\frac{\partial E^{\lambda=0}}{\partial W} = \epsilon_{l+1} \frac{\partial f(W_l x_l)}{\partial W_l} x_l^T = 0 \tag{17}$$

which results in the CHL update in PC simply equalling the gradient at the clamped equilibrium,

$$\Delta W_{CHL}^{PC} = \frac{1}{\lambda} \Big[ \frac{\partial E^\lambda}{\partial W} - \frac{\partial E^{\lambda=0}}{\partial W} \Big] = \frac{1}{\lambda} \frac{\partial E^\lambda}{\partial W} \tag{18}$$

since the free phase Hebbian weight update in PCNs is $0$. This means that only a single clamped phase is required to train PCNs instead of two for the Hopfield energy. This results in significant computational savings as well as increased biological plausibility since the gradients of the free phase do not need to be stored in the circuit during convergence to the clamped phase equilibrium.

### 3.4 EP AS INFINITESIMAL PERTURBATION

While CHL algorithms typically clamp the output of the network to the targets and hence utilize a large output $\lambda$, necessitating that feedback through the network be weak, EP takes the opposite approach and explicitly utilizes a small $\lambda$ directly. EP can be derived by essentially taking an infinitesimal perturbation of $\lambda$ and comparing the difference between the two phases – the free phase, and an infinitesimally 'nudged' phase where $\lambda \to 0$. To derive EP, if we take the derivative of the decomposition of the energy in Equation 9 with respect to $\lambda$, we can write the supervised loss $\mathcal{L}$ as,.

$$E = \mathcal{I} + \lambda \mathcal{L} \implies \mathcal{L} = \frac{dE}{d\lambda} \tag{19}$$

We wish to compute the gradient of the supervised loss with respect to the parameters $W$,

$$\frac{d\mathcal{L}}{dW} = \frac{\partial}{\partial W} \Big[ \frac{\partial E}{\partial \lambda} + \frac{\partial E}{\partial x} \frac{\partial x}{\partial \lambda} \Big] = \frac{\partial}{\partial W} \frac{\partial E}{\partial \lambda} = \frac{\partial}{\partial \lambda} \frac{\partial E}{\partial W} \tag{20}$$

where firstly we have utilized the fact that in the free phase equilibrium $\frac{\partial E}{\partial x} = 0$ and we have substituted in Equation 19. In the final step we have utilized the fact that partial derivatives commute. We have thus expressed the gradient of the loss in terms of a derivative with respect to the supervised loss scale term $\lambda$. One way to compute this derivative is by finite differences, thus for a small $\lambda$,

$$\frac{\partial L}{\partial W} = \lim_{\lambda \to 0} \frac{1}{\lambda} \Big[ \frac{\partial E^\lambda}{\partial W} - \frac{\partial E^{\lambda=0}}{\partial W} \Big] \tag{21}$$

Where the first gradient is taken at the nudged phase equilibrium and the second at the free phase equilibrium where $\lambda = 0$. By the standard properties of finite difference approximations, this update rule becomes exact as $\lambda \to 0$. This recovers the EP weight update rule. Importantly, we do not make any assumptions about the form of the energy function. This means that this method is general for any energy function and not just the Hebbian energy used in (Scellier & Bengio, 2017). For instance, instead the PC squared energy function could be used which would derive an additional algorithm that approximates BP in the limit that has not yet been proposed in the literature – although it would be closely related to existing results (Whittington & Bogacz, 2017). We implement and investigate this algorithm below and demonstrate that its convergence to BP properties match precisely those predicted by our theory.

## 4 EXPERIMENTS

Here, we use our theoretical framework to derive a novel BP-approximating variation which we call 'PC-Nudge' that appproximates BP through the use of weak supervision instead of the CHL-like weak-feedback limit previously reported in (Whittington & Bogacz, 2017). Instead of adjusting the precision ratio, PC-nudge instead varies the weighting of the supervised loss with an EP-like coeffi-

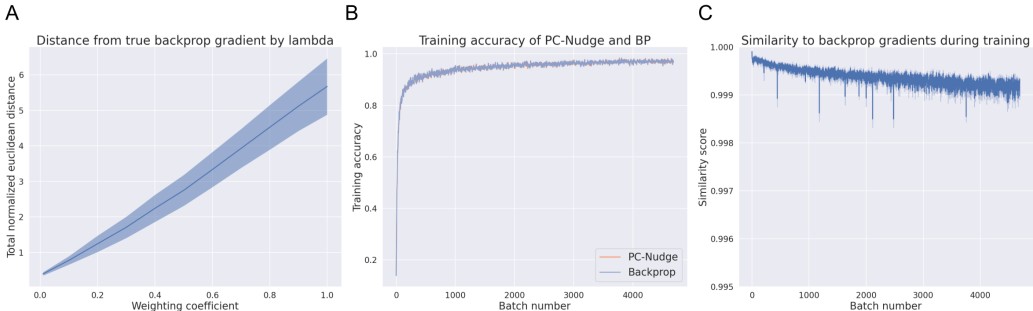

Figure 3: **A**: A linear relationship between the $\lambda$ weighting coefficient and the approximation of the PC-Nudge algorithm to BP was robustly observed at initialization. Error bars represent standard deviations across 20 random initializations. **B**: The PC-Nudge network trained with almost identical accuracy across batches as the corresponding ANN. **C**: Cosine similarity during training between BP gradients and those estimated by PC-Nudge. Although similarity appears to decline somewhat, it never goes below $0.999$ and has no observable impact on training. Results were obtained with a 4-layer MLP with relu activation functions trained with a predictive coding quadratic energy function.

cient $\lambda$. PC-Nudge can be defined as a gradient descent on the following free energy functional,

$$E_{\text{PC-Nudge}} = \lambda \epsilon_L + \sum_{l=1}^{L-1} \epsilon_l^2 \tag{22}$$

Which results in a standard PCN except with a downweighted final layer since $\lambda$ is defined to be small. To counteract the downscaling of the supervised loss, all learning rates in the network are rescaled by $\lambda$ with a new learning rate $\tilde{\eta} = \frac{\eta}{\lambda}$ resulting in the same training rate as BP. As a demonstration experiment, in Figure 3, we compared PC-Nudge to BP on training a simple 4-layer MLP with relu activation function on the MNIST dataset. The actual scale of the experiment is largely irrelevant since as $\lambda \to 0$, the gradients computed by PC-Nudge converge towards the exact BP gradient and hence we expect training performance to be the same for any architecture and scale. In practice, for our network, we found a $\lambda = 0.01$ to achieve extremely close convergence to BP throughout all of training and correspondingly identical training performance. We plot the training accuracies of the PC-Nudge and backprop algorithms (which are almost identical) in panel B, and the cosine similarity between the PC-Nudge weight updates and gradients in panel C. We found that this similarity decreases marginally throughout training (but always remained high at above $0.999$) but not enough to affect training in a noticeable way. We conjecture this occurs because the norm of the BP gradients decreases during training, thus necessitating a smaller $\lambda$ to maintain equivalent scale. Finally, in Figure 3A, we also found that the distance between the PC-Nudge and BP gradients at initialization was approximately a linear function of $\lambda$ over a wide range of sensible $\lambda$ values. Detailed information regarding our implementation and experiments can be found in Appendix A.2 and an explicit algorithm for PC-Nudge can be found in Appendix A.2.3.

## 5 DISCUSSION

In this paper, we have proposed a novel and elegant theoretical framework which unifies almost all the disparate results in the literature concerning the conditions under which various 'biologically plausible' algorithms approximate the BP of error algorithm. We have demonstrated that these convergence results arise from a shared and fundamental mathematical property of EBMs in the infinitesimal inference limit. This limit is essentially when the clamped phase equilibrium of the EBM converges towards its initial free phase equilibrium. We have shown that the approximation error scales linearly with the distance between the free and clamped equilibria. This limit can be achieved in one of two ways – either by enforcing weak feedback through the network, leading to the CHL algorithm of (Xie & Seung, 2003) and the PC result of (Whittington & Bogacz, 2017), or else by scaling down the contribution of the supervised loss to the energy function, by which one obtains EP (Scellier & Bengio, 2017) and our novel PC-Nudge algorithm.

While the analysis in this paper covers almost all of the known results concerning approximation or convergence to BP, there exist several other families of biologically plausible learning algorithms which do not converge to BP but which nevertheless can successfully train deep networks. These include target-propagation (Lee et al., 2015; Meulemans et al., 2020), the prospective configuration algorithm in PCNs networks Song et al. (2022), and the feedback alignment algorithm and its variants (Lillicrap et al., 2016; Nøkland, 2016; Launay et al., 2020). Whether these algorithms fit into an even broader family or can be derived as exploiting other mathematical properties of a shared EBM structure remains to be elucidated. Moreover, our mathematical results rely heavily on the relatively crude machinery of Taylor expansions around the clamped phase equilibrium. An interesting avenue for future work would be to apply more sophisticated methods to obtain better approximations and understanding as to the scaling of the approximation error as the distance between the clamped and free phase equilibria increases.

## ACKNOWLEDEGMENTS

This work has been supported by BBSRC grant BB/S006338/1 and MRC grant MC_UU_00003/1. Thomas Lukasiewicz has been supported by the Alan Turing Institute under the UK EPSRC grant EP/N510129/1 and by the AXA Research Fund.

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

# A  APPENDIX

## A.1

Related Work The main contribution of this paper is in tying together many previous threads in the literature and presenting a unified view of algorithms approximating BP. As such many of the ideas and derivations here are present, albeit in isolation, in previous literature. A similar relationship between the gradients of the supervised loss and the dynamics of neural activity at free phase equilibrium has been observed before in (Bengio & Fischer, 2015). The single-step update is derived in (Song et al., 2020) although the derivation is specific to PC and was not conceptualized as a general property of energy based models. The derivation of EP as a finite difference approximation of the gradient of the $\lambda$ parameter follows that in (Scellier & Bengio, 2017). A relationship between the PC approximation result of (Whittington & Bogacz, 2017) and EP was hinted at in (Whittington & Bogacz, 2019) but not explicitly worked out, and we argue that CHL is actually a better fit for PC since the output is fully clamped and not 'nudged' as in EP. The idea of CHL being derived as a gradient descent on a contrastive loss function equal to the difference between the free and clamped equilibrium is an explicit part of the CHL literature (Movellan, 1991; Xie & Seung, 2003) although to our knowledge our derivation of this loss function as a consequence of a general energy based model is novel. Similarly, our derivation in Section 3.3 expressing the infinitesimal inference limit in its full generality is novel as is explicit working out of the approximation terms.

## A.2 EXPERIMENTAL DETAILS

All experiments used the MNIST dataset. A standard training-test split of 50,000 train images and 10,000 test images was used. All MNIST images were normalized so that their pixel values lay in the range [0,1].

All experiments were undertaken on a 3-layer MLP architecture. The MLP consisted of three layers of size 784,128,64,10 with a one-hot output vector of the correct class. A relu activation function was used with a softmax activation at the output along with the cross-entropy loss function. The network was initialized with using Pytorch's default (Xavier) initialization for linear layers.

The inference phase of the PCN was accomplished by explicitly integrating Equation 4 for 50 iterations. An inference learning rate of 0.1 was used for the inference phase. All precision parameters $\Pi_l$ were implicitly set to the identity matrix.

All code required to reproduce all experiments and figures in this paper will be made publically available at `github_address`.

### A.2.1 FIGURE 2 IMPLEMENTATION DETAILS

In Figure 2A, the energy was computed as the PC energy of the sum of squared prediction errors at each timestep of inference. The internal energy was defined as the sum of the squared prediction errors for the first two layers and the supervised backprop loss was the prediction error of the final layer. This lets us interpret the network as effectively redistributing prediction errors from the output layer into the internals of the network, but doing so in a way that allows the total norm of the prediction errors to decrease.

In Figure 2B, we computed the Euclidean distance between the true backprop gradients and that approximated by the EBM during inference. We used an inference step-size of 0.1. We computed our results on a randomly initialized 3-layer MLP as described above or, equivalently, a PC-Nudge with $\lambda = 1$. Error bars represent the standard deviation across 10 random initializations.

In Figure 2C, we investigated the relationship between the approximation error to the true backprop gradients and that computed by the EBM (in our case a 3-layer MLP PCN). Estimated gradients were computed by our PC-Nudge algorithm where distances between the free and clamped phase were varied in practice by testing different $\lambda$ parameters. The total euclidean distance $d(x, y) = \sqrt{\sum_i (x_i - y_i)^2}$ between both free phase (forward pass) and clamped phase activations as well as between the BP and PC-Nudge estimated gradients were used. The equilibrium distances were summed across all layers while the gradient distances were summed across all parameters (weights and biases) in the network. Since the exact value of the distances is largely irrelevant we simply summed and did not normalize either the gradients or equilibrium distances so in practice the exact values of these distances scale with layer-width and depth of the network. For this figure we chose 10 represenentative examples of the same input image (a randomly selected mnist image and label) across initializations of the network.

### A.2.2 FIGURE 3 IMPLEMENTATION DETAILS

The learning results were obtained with the SGD with momentum optimizer with a learning rate of 0.001 for both BP and PC-nudge with learning rate of $\frac{0.001}{\lambda}$. The momentum parameter was set to 0.9. A minibatch size of 64 was used throughout. $\lambda$ was set to 0.001.

Panel A was produced by computing the Euclidean distance between the BP and PC-Nudge gradients after 50 inference steps for various values of $\lambda$. The shaded area represents the standard deviation over 10 seeds. Panel B shows the batch-by-batch training accuracies of the BP and PC-Nudge network superimposed. The gradients computed by the two approaches were so similar that the weight and gradient updates ended up being almost identical. Panel C shows the cosine similarity (or normalized dot product) between the PC-Nudge estimated gradients and the true BP gradients. We computed the similarity as,

$$\text{similarity}(x, y) = \frac{x \cdot y}{||x||_2 ||y||_2} \tag{23}$$

A similarity score of 1 is the highest possible score and $-1$ is the lowest.

### A.2.3 PC-NUDGE ALGORITHM

---

**Algorithm 1** PC-Nudge algorithm

---

**Data:** Dataset $\mathcal{D} = \{d_i, l_i\}$, set of neural activities $\{x_l\}$, set of weights $\{W_l\}$ inference learning rate $\eta_x$, weight learning rate $\eta_\theta$, 'nudge' parameter $\lambda$

**begin**

    /* For each minibatch in the dataset                                      */

    **for** $(d, l) \in \mathcal{D}$ **do**

        /* Fix input layer to data                                           */

        $x_0 \leftarrow d$

            /* Forward pass to compute predictions                       */

        **for** $x_l \in$ *network layers* $\{l\}$ **do**

            $x_l \leftarrow f(W_l x_{l-1})$

        /* Compute output error                                           */

        $\epsilon_L \leftarrow \frac{1}{\lambda}(T - x_L)$

            /* Inference phase                                           */

        **while** *not converged* **do**

            **for** $(x_l, \epsilon_l) \in$ *network layers* $\{l\}$ **do**

                /* Compute prediction errors                            */

                $\epsilon_l \leftarrow x_l - f(W - l x_{l-1})$

                    /* Update neural activities                          */

                $x_l^{t+1} \leftarrow x_l^t + \frac{\eta_x}{\lambda}\frac{\partial E}{\partial x_l^t}$

        /* Update weights at equilibrium                                 */

        **for** $W_l \in$ *network weights* $\{W_l\}$ **do**

            $W_l^{t+1} \leftarrow W_l^t + \frac{\eta_\theta}{\lambda}\frac{\partial E}{\partial W_l^t}$

---

## A.3   BIOLOGICAL PLAUSIBILITY OF EBMS

In the literature, the EBMs described in this paper, are often motivated and described as biologically plausible, unlike backprop. Due to the fact that we have shown that all of these algorithms approximate backprop in the limit, it is important to consider how this affects the claimed biological plausibility of these algorithms. For the first-step algorithms we describe (section 3.2), these algorithms are exactly equivalent to backprop (up to a scaling factor) and hence have identical biological plausibility properties as backprop. In effect, these first step algorithms simply propose an additional mathematical interpretation of backprop as taking a single step towards the minimum of some energy function, thus allowing us to situate backprop as a special case of a more general EBM framework.

EBMs are often claimed to be biologically plausible in the literature due to their properties of highly parallel and local updates in the inference phase. This is due to the typically layerwise form of the energy functions used to derive the dynamics of neural activities in the EBM. This means that, in theory, the neurons in an EBM can simultaneously update themselves in parallel, responding only to their neighbours in the layers above and below. Information propagates through the network at the same layerwise rate as backprop, however. As such, the general EBMs have some claim to be eliminate some of the biological implausibilities of backprop such as its sequential forward and backward sweeps where each layer must 'wait' for previous layers to complete either the forward or backward pass. EBMs do not address other claimed biological implausibilities such as the weight transport problem (Lillicrap et al., 2016; Crick et al., 1989) and also introduce new ones such as how information from the free phase is stored during the clamped phase, and how the two phases can be coordinated.

Additionally, the infinitesimal inference limit required to closely approximate backprop typically leads to biologically implausible constraints such as extremely weak feedback connections (when feedback connections in the brain are strong) or an extremely small $\lambda$ parameter value leading to tiny differences in values between phases which must be stored and then subtracted accurately while computation in the brain likely takes place in relatively limited precision.

However, regardless of the biological plausibility of these EBM limits, we believe that our work is important since it lets us understand that backprop emerges as a general limit of EBMs and how to unify existing EBMs in the literature into a single, extremely simple framework.

