# OpenReview forum: "Backpropagation at the Infinitesimal Inference Limit of Energy-Based Models: Unifying Predictive Coding, Equilibrium Propagation, and Contrastive Hebbian Learning"
_ICLR.cc/2023/Conference — ICLR 2023 poster_

### Official Review · Reviewer_oU1M · 2022-10-24

**Confidence:** 4
**Clarity, Quality, Novelty And Reproducibility:** The paper is clear, high-quality, and…
**Correctness:** 4
**Technical Novelty And Significance:** 2
**Empirical Novelty And Significance:** 2
**Recommendation:** 6

**Strength And Weaknesses:**

Strengths:
- The paper is well-written and clear.
- The new energy-based model perspective is a way to compare and contrast three different non-backprop  learning algorithms, and understand how they deviate from backprop (gradient descent on the supervised loss).

Weaknesses:
- It is not obvious to me that viewing these algorithms as gradient descent on an energy function is useful.
- One justification given for proposing this new framework is that it can be used to propose new algorithms.  PC-Nudge is proposed. However, this algorithm seems to be a minor variation on existing algorithms, and the experimental results on MNIST are not particularly interesting.
- Performing gradient descent on an energy function is not necessarily biologically plausible, and this paper does not have much to say about the biological plausibility of these various algorithms.

**Summary Of The Paper:**

Three types of biologically-plausible learning algorithms --- predictive coding, contrastive hebbian learning, and equilibrium propagation --- are reformulated as energy based models. This framework is used to provide a unifying description for when these algorithms closely approximate gradient descent (backprop), and the authors use this new perspective to suggest a new variant of predictive coding called PC-Nudge. Experiments with a multi-layer neural network on MNIST are used to support the proposed ideas.

**Summary Of The Review:**

This is a well-written paper that might be of interest to some in the community, but despite my familiarity with the algorithms discussed, I was unable to see how it is particularly useful to view them from the energy-based model perspective.

---

> ### Author Response · Authors · 2022-11-15
> **Response to reviewer oU1M**
>
>  We thank the reviewer for their clear and straightforward review. The main criticism of this reviewer is not technical but primarily about the importance and utility of the results we present, which is ultimately a matter of intellectual interest and taste.
>
> We believe that the primary utility of our results is due to their theoretical unification of the disparate algorithms in the literature through a simple and easy to understand framework. This makes both understanding the strengths and the weaknesses of current approaches much easier and reveals that they are all much more related than was previously thought.
>
> Specifically, our results show that all of the ‘biologically plausible’ algorithms that have been shown to closely approximate backprop all follow and can be derived from a given limit and moreover exactly why and how this limit works. We believe that this is a substantial advance upon prior work which, from our perspective, was a number of seemingly disparate and unrelated algorithms and results. The underlying question of whether and how backprop can be implemented or approximated in a local and parallel way is a highly important one in both machine learning and neuroscience and while we don’t come close to definitively answering that here, we do provide a much needed theoretical unity to current work which can then be built upon moving forward. In some sense, since the limit is fairly straightforward and deflationary, this is a negative result, but this is important for the field to know.
>
> In terms of the specific arguments of the reviewer:
>
> 1.) All of these algorithms are usually explicitly derived from a gradient descent on an energy function and can be most easily analyzed and understood in this way. Their parallelizability properties ultimately arise from the decomposability of the energy function into ‘local’ layerwise parts, and moreover the energy function approach lets us easily define the limits where they come to approximate backprop, as we do in this paper.
>
> 2.) Indeed the PC-Nudge algorithm is not particularly interesting. Our aim was simply to show that our limit is constructive in the sense that it is straightforward to derive new algorithms that reach it and can approximate backprop with arbitrary fidelity. The experimental results on MNIST only serve to validate that the algorithm does converge to backprop and it can be made to do so for any network architecture or dataset.
>
> 3.) We agree that performing gradient descent on an energy function is not necessarily biologically plausible, and also that new ideas are probably needed in this space. However, this was not the goal of the paper, which is to unify existing algorithms under a single simple theoretical framework, and all current algorithms are phrased in terms of gradient descent on an energy function.

---

> > ### Comment · Reviewer_oU1M · 2022-11-19
> > **Raising my score**
> >
> > My main concern was whether the new perspective provided by this paper is of interest to the community. Since the other reviewers find it to be a valuable contribution, I am raising my score.

---

### Official Review · Reviewer_wqJn · 2022-10-25

**Confidence:** 4
**Clarity, Quality, Novelty And Reproducibility:** Overall, this is a well-written paper…
**Correctness:** 4
**Technical Novelty And Significance:** 3
**Empirical Novelty And Significance:** Not applicable
**Recommendation:** 8

**Strength And Weaknesses:**

Strengths:
Analysis of the small $\lambda$ limit
The PC-nudge model
Discussion of contrastive Hebbian learning, starting from PC.

Minor issues:
In equations (3) and (4) the indexing conventions of $W_l$ and $x_l$ seems to be inconsistent with the definition of $\epsilon_l$ in the previous page.
Under equation(5), the sentence mentioning “..neurons which are co-active at the clamped phase are strengthened…” is confusing. What does 'neurons being strengthened' mean? What is really strengthened? Synaptic strengths? That seems to have the opposite effect. A clearer explanation here would help.


**Summary Of The Paper:**

This paper starts from a PCN and expands around the limit where the output is not tethered to the target. Perturbing in the parameter $\lambda$, which may be interpreted as the precision parameter deciding how tightly would the target be tied to the output, builds a useful bridge to many other approaches to training neural networks.

**Summary Of The Review:**

 I found this paper useful for relating predictive coding networks with other competing approaches. It is a welcome addition to the literature.

---

> ### Author Response · Authors · 2022-11-15
> **Response to reviewer wqJn**
>
> We thank the reviewer for their helpful review and bringing our attention to various minor issues.  Indeed, the indexing conventions were not consistent and we have fixed this in the updated version. Secondly, we have adjusted the wording around how CHL works.

---

### Official Review · Reviewer_mXjX · 2022-10-25

**Confidence:** 4
**Correctness:** 3
**Technical Novelty And Significance:** 3
**Empirical Novelty And Significance:** 2
**Recommendation:** 6

**Clarity, Quality, Novelty And Reproducibility:**

### Clarity

Overall, the paper is well-written. Some minor comments:

Both figures (especially Fig. 1) should be exported in either a vector format or much higher resolution.

Fig.3B: PC-nudge is completely hidden behind backprop dynamics.

Sec. 2.3, first line: capitalize Hebbian.

Before Eq. 15: If we assume that the gradients of each component of the energy with respect to the weights -is- continuous and differentiable -> are.

Just before Sec. 3.4: capitalize Hopfield.

Eq. 23: the norms should not be squared in cosine similarity.

### Quality

The technical contributions are somewhat limited as the paper mainly re-writes the three existing algorithms in a different form. The experimental contributions are very limited as the proposed algorithm, PC-nudge, is tested on a single network and only on MNIST. It's not clear if the reasonable alignment with backprop will hold for large networks/harder tasks. Also, what's the takeaway from this algorithm?

### Novelty

The connection between different algorithms is, to my knowledge, novel. All relevant literature is cited.

### Reproducibility

The code is not provided but it's implied that it will be. The authors can attach a zip file with the code to the submission.


**UPDATE**: the code is now provided.

**Strength And Weaknesses:**

### Strengths

The paper unifies several backprop approximation under the same energy-based framework.

This framework leads to different approximations.

### Weaknesses

I don’t understand what the take-home message is. Predictive coding and equilibrium propagation were already formulated in a similar fashion, but what can we do with it now? I'd guess deriving more plausible algorithms would be interesting, but it's not really discussed.

Contrastive Hebbian learning narrowly fits the proposed framework. Eq. 18 seems to be just the gradient of the loss (since $I$ is zero in that case), which is not what CHL is doing.




**Summary Of The Paper:**

The paper unifies several approximation to backpropagation through the energy-based models view.

**Summary Of The Review:**

Potentially interesting connection between algorithms, but I'm not sure what we can do with it. I'm willing to discuss it further, however.

**UPDATE**: updated the score from 5 to 6 post-rebuttal.

---

> ### Author Response · Authors · 2022-11-15
> **Response to reviewer mXjX**
>
> We thank the reviewer for their detailed comments and clear effort put into reading and understanding the paper. The primary concerns of the reviewer appear to be about the technical utility and degree of contribution of our work.
>
> Regarding the technical contribution: our contribution in this paper is to provide a unified theoretical framework to understand three major algorithms in the literature which have not been unified before, as well as to provide a single unified limit which encompasses multiple results in the literature demonstrating the close approximation between these algorithms and backprop. While our framework is very simple, we believe that this is a virtue and increases the value of our work, since it is novel and makes it extremely clear and obvious how these algorithms are related while this has not been at all obvious in the prior literature.
>
> > The experimental contributions are very limited as the proposed algorithm, PC-nudge, is tested on a single network and only on MNIST. It's not clear if the reasonable alignment with backprop will hold for large networks/harder tasks.
>
> The PC Nudge algorithm is intended to be a constructive proof that our limit works and can be used to derive new algorithms. While we only tested on MNIST, mathematically, it is clear that this algorithm would still approximate backprop for larger networks or harder tasks, since we can approximate backprop with arbitrary fidelity by adjusting the $\lambda$ parameter. Of course at some point this may be obstructed in practice by numerical error. Other works have shown that PC and EP can approximate backprop extremely well on much larger networks and harder tasks and there is no reason this would not also work for PC-Nudge.
>
> As to your minor comments – these corrections have been helpful in improving the manuscript and we have fixed them in the updated version. We have also attached the code in a zip file – apologies for forgetting this in the original submission

---

> > ### Comment · Reviewer_mXjX · 2022-11-15
> > **Still have concerns but updated the score to 6**
> >
> > Thank you for the response!
> >
> > Your comments make sense, but I still have the aforementioned concerns regarding the take-home message of this work. So, I'm still considering the submission to be borderline, but I think it's interesting enough (+ the code is now there) so I'm updating the score to 6.

---

### Official Review · Reviewer_Cz4e · 2022-10-25

**Confidence:** 4
**Correctness:** 4
**Technical Novelty And Significance:** 3
**Empirical Novelty And Significance:** 3
**Recommendation:** 8

**Clarity, Quality, Novelty And Reproducibility:**

The novelty of the present work stems from the amalgamation of several lines of work under one framework.  The manuscript is fairly clear and staightforward

**Strength And Weaknesses:**

Strengths

1. The authors gather previously disparate theoretical results together under the same umbrella of energy-based models.

2. Several versions of predictive coding, along with equilibrium propagation and contrastive Hebbian learning are shown how, under the same framework, they can be derived as approximations to backprop.

3. A novel algorithm, "PC-nudge," is shown to be derivable from the proposed framework

Weaknesses

No major weaknesses of the manuscript were found



**Summary Of The Paper:**

A number of different algorithms have been proposed as more biologically plausible alternatives to backpropagation.  While different works have shown how several of these proposals approximate backprop by converging to it in some limit, the authors here outline an overall framework of energy-based models that encompasses several predictive coding proposals, equilibrium propagation, and contrastive Hebbian learning, showing how they all can be formulated as approximations to backprop.  By breaking the global energy into components corresponding to the internal energy of the network and the network's cost function, the authors show that different algorithms can approximate backprop by finding equilibrium points near the minimum of the internal energy (by, e.g., employing weak feedback).  They finally show that other algortihms can be derived from their approximation, with "PC-nudge" described as an example.

**Summary Of The Review:**

Several results are gathered and combined by showing how a number of proposed biologically plausible alternatives to backprop can be shown to approximate backprop when formulated as energy-based models. The work then leads to an example of a novel algorithm that can be derived using this common framework

---

> ### Author Response · Authors · 2022-11-15
> **Response to reviewer Cz4e**
>
> We thank the reviewer for their detailed and positive review.

---

### Decision · Program_Chairs · 2023-01-20

**Decision:**

Accept: poster

**Justification For Why Not Higher Score:**

Even though the current version of the paper is good for publication, but it is not perfect and the current experiments for justification can be further improved and strengthened.

**Justification For Why Not Lower Score:**

The paper provides a novel theoretical framework that relates predictive coding, equilibrium propagation and contrastive Hebbian learning. All reviewers lean to accept it.

**Metareview: Summary, Strengths And Weaknesses:**

This paper proposes a theoretical framework which unifies all the disparate results in the literature concerning the conditions under which various ‘biologically plausible’ algorithms approximate the BP of error algorithm. The authors demonstrate that these convergence results arise from a shared and fundamental mathematical property of EBMs in the infinitesimal inference limit. The paper validates the proposed framework by deriving a new PC Nudge algorithm and testing it on the MNIST dataset. After the rebuttal, all 4 reviewers lean to accept the paper (particularly, 2 of them champion the paper, with each rating a score 8: accept) because the paper is interesting, novel, and makes a valuable contribution for relating predictive coding, equilibrium propagation and contrastive Hebbian learning. The rebuttal has addressed all the major concerns raised by reviewers. The AC agrees with the judgements of the reviewers and recommends accepting the paper.


**Note From Pc:**

if the above contains the word "oral" or "spotlight" please see: "oral" presentation means -> notable-top-5% and "spotlight" means -> notable-top-25%. As stated in our emails, we are disassociating presentation type from AC recommendations